# Impact of Actively Offering Influenza Vaccination to Frail People during Hospitalisation: A Pilot Study in Italy

**DOI:** 10.3390/vaccines11121829

**Published:** 2023-12-08

**Authors:** Alessandra Fallucca, Patrizia Ferro, Luca Mazzeo, Luigi Zagra, Elena Cocciola, Roberta Oliveri, Antonino Tuttolomondo, Alida Benfante, Salvatore Battaglia, Nicola Scichilone, Nicola Veronese, Marco Affronti, Mario Barbagallo, Alessandra Casuccio, Francesco Vitale, Vincenzo Restivo

**Affiliations:** 1Department of Health Promotion, Mother and Child Care, Internal Medicine and Medical Specialties, University of Palermo, 90133 Palermo, Italy; patrizia.ferro@unipa.it (P.F.); luca.mazzeo@unipa.it (L.M.); luigi.zagra@unipa.it (L.Z.); alessandra.casuccio@unipa.it (A.C.); francesco.vitale@unipa.it (F.V.); 2Internal Medicine and Stroke Care Ward, Department of Health Promotion, Maternal and Infant Care, Internal Medicine and Medical Specialties “G. 6 D’Alessandro”, University of Palermo, 90133 Palermo, Italy; elena.cocciola@unipa.it (E.C.); robertaoliveri_1995@yahoo.it (R.O.); antonino.tuttolomondo@unipa.it (A.T.); 3Division of Respiratory Diseases, Department of Health Promotion Sciences, Maternal and Infant Care, Internal Medicine and Medical Specialties (PROMISE), University of Palermo, 90133 Palermo, Italy; alida.benfante@unipa.it (A.B.); salvatore.battaglia@unipa.it (S.B.); nicola.scichilone@unipa.it (N.S.); 4Geriatric Unit, Department of Medicine, University of Palermo, 90133 Palermo, Italy; nicola.veronese@unipa.it (N.V.); mario.barbagallo@unipa.it (M.B.); 5Internal Medicine Unit, AOU Paolo Giaccone Policlinic, General Hospital, 90133 Palermo, Italy; marco.affronti@policlinico.pa.it; 6School of Medicine, University Kore of Enna, 94100 Enna, Italy; vincenzo.restivo@unikore.it

**Keywords:** influenza vaccination, vaccination offer, frail people

## Abstract

Despite the worldwide recommendations for influenza immunisation, vaccination coverage for patients exposed to the highest risk of severe complications is still far from the optimal target. The need to take advantage of alternative methods to provide vaccination is essential. This study presents a hospital-based strategy which offers influenza vaccination to inpatients at discharge. This study was conducted during the 2022–2023 influenza season at the University Hospital of Palermo. A questionnaire was administered to identify the determinants for the acceptance of influenza vaccination in the frail population. Overall, 248 hospitalised patients were enrolled, of which 56.1% were female and 52.0% were over 65 years of age. The proportion of patients vaccinated against influenza during hospitalisation was 62.5%, an increase of 16% in influenza vaccination uptake among frail people in comparison with the previous influenza season (46.8% vaccinated during the 2021–22 influenza season). Factors significantly associated with vaccination acceptance were the following: to have received influenza vaccine advice from hospital healthcare workers (OR = 3.57, *p* = 0.001), to have been previously vaccinated for influenza (OR = 3.16 *p* = 0.005), and to have had a low level of education (OR = 3.56, *p* = 0.014). This study showed that offering influenza vaccination to hospitalised patients could be an effective strategy to increase vaccination coverage in the most vulnerable population, and these findings could be useful for planning and improving future influenza vaccination campaigns.

## 1. Introduction

Due to influenza’s wide spread and contagiousness, influenza epidemics represent an important public health problem causing, every year, thousands of cases, and are a significant source of costs because of the management of cases and complications of the disease and the implementation of control measures [1]. The European Center for Disease Control estimates that up to 70,000 deaths due to influenza-related causes occur in Europe each year [2]. In particular, pneumonia associated with influenza virus infection is reported to be among the top ten causes of death in Italy, with the majority of deaths occurring among frail and immunocompromised people with comorbidities and elderly people aged over 65 years old [3,4,5].

Vaccination is the most effective measure to prevent more severe complications and it is strongly recommended for susceptible populations [6,7]. According to World Health Organization guidelines, a threshold of 75% vaccination coverage against influenza is necessary among the general population, as a minimum achievable objective, and a threshold of 95% among the frail population is necessary to prevent possible negative outcomes and reduce the morbidity related to seasonal influenza [1,8]. Despite the recommendations and the free offer of the vaccine, in Italy, about 20 doses were administered for every 100 inhabitants during the 2021–2022 influenza season and a vaccination coverage of 58% was achieved among people over 65 [9].

Low influenza vaccine uptake rates among specific risk groups contribute to the burden of disease and remain a major public health challenge. Most vaccinations for adults and elderly people are administered through primary care. However, the low coverage highlights that relying exclusively on traditional primary care for vaccination might not result in high vaccination coverage [10]. The effectiveness of vaccine catch-up interventions in increasing vaccination adherence has been demonstrated and there are several strategies that can be adopted, such as active calls to remember vaccination, educational interventions to promote immunisation and vaccinations in healthcare settings other than primary care facilities [11].

In Italy, in addition to the national recommendations for vaccination issued by the Ministry of Health, the administrative regions can implement additional health strategies to achieve the health goals for their population [1]. At the beginning of the 2022–2023 influenza vaccination campaign, the Sicilian Health Authority published a decree to vaccinate all categories at risk of complications by inviting all hospitals, nursing homes and healthcare facilities to offer influenza vaccinations to eligible inpatients before discharge [12]. 

Although the influenza vaccine is offered free of charge to at-risk people, adherence to vaccination is also threatened by some contextual, socio-demographic and physical barriers [13]. Lack of confidence in vaccines, lack of adequate information about the complications due to infection and a poor attitude to a healthy lifestyle are some of the main factors related to influenza vaccine hesitancy [13,14,15]. The literature is lacking in studies that have evaluated the reasons for refusing and accepting an influenza vaccination specific to populations at risk of complications, especially over-65-year-olds and chronically ill individuals. The main aim of this study was to evaluate the impact of an innovative vaccine-offer strategy in a hospital setting and then to analyse the facilitators and barriers associated with influenza vaccination uptake among frail people.

## 2. Materials and Methods

This cross-sectional study was conducted from November 2022 to February 2023 at the University Hospital of Palermo Policlinico P. Giaccone to explore the factors associated with the acceptance of influenza vaccination and the impact of actively offering influenza vaccination to hospitalised patients. The 2022/2023 influenza vaccination campaign was conducted in hospital wards through the administration of illustrative brochures, posters and communications about the risks and complications of influenza infection and the recommendations of vaccination for frail people. A team of vaccinating physicians from the Department of Hygiene and Preventive Medicine of the University of Palermo offered vaccination three times a week to the hospital clinic and surgery patients. Collaboration with the medical and nursing staff and consultation of the medical records allowed the identification of the individuals eligible for vaccination. Influenza vaccination was offered to frail patients, i.e., those over 65 years old or affected by chronic clinical conditions or comorbidities, who were the main target for vaccination as stipulated by ministerial recommendations for the influenza campaign. The vaccine was offered directly during discharge from the hospital, after informing the patient about the risks and benefits of the influenza vaccine and after assessing the patient’s clinical condition as suitable to receive the vaccination (absence of inflammation and fever) [1].

A validated and structured questionnaire was administered to all patients eligible for vaccination. The questionnaire was addressed to the patients with the aim of identifying factors associated with the uptake of vaccination against influenza. In accordance with the pre-existing literature, the following items were investigated: personal and socio-demographic data, such as age, gender, work activity and level of study; health habits and behaviour, for instance, smoking and eating habits; state of health and comorbidities; previous influenza vaccination; and physician recommendations [16,17]. The variable “educational level” was categorised as follows: “Low” for primary school qualification, “Medium” for secondary school qualification and “High” for high school diploma and degree [18]. The variable “Economic status” was arbitrarily categorised based on self-reported data from the interviews: patients who declared reaching the end of the month very easily or quite easily were, respectively, classified as “High” or “Medium-High” level; patients who declared having some or many economic difficulties were classified as “Medium-low” and “Low” level, respectively.

This study was approved by the Ethical Committee Palermo 1 at the meeting in October 2022 (09/2022). 

Patient information was transferred from paper formats to computer files; a database was constructed to record data using Excel—Office 2021. All collected data were analysed using Stata/SE 14.2 statistical software (Copyright 1985–2015, StataCorp LLC, 4905 Lakeway Drive, College Station, TX 77845, USA. Revision 29 January 2018). The normality of the distribution for the quantitative variables was assessed using the Skewness and Kurtosis test. Mean and standard deviation (SD) were chosen for reporting normally distributed variables, whereas median and interquartile range (IQR) were used for non-normal distribution. The frequencies, absolute and relative, were calculated for the qualitative variables. Student’s *t*-test and Wilcoxon test were used to evaluate the normal or non-normal distribution of quantitative variables, whereas for the qualitative variables, a Chi2 test was performed. Univariable logistic regression analysis was performed to evaluate the factors associated with influenza vaccination acceptance. A multivariable logistic regression model was built to analyse factors found to be associated with a *p*-value lower or equal than 0.05 through the univariable analysis. Moreover, a priori confounding variables, such as age and sex, were included in the multivariable model. For all analyses, a *p*-value of 0.05 was assumed to be statistically significant. 

## 3. Results

Overall, 253 patients hospitalised during the 2022–2023 influenza season were enrolled, but 5 people denied consent to be interviewed (response rate 98%). Approximately half of the 248 people interviewed were female (56.1%, n = 139) and approximately half were over 65 years old (52.0%, n = 129). The majority of the respondents declared not having economic difficulties, defining their economic status as “medium-high” (37.1%, n = 92) or “medium-low” (39.1%, n = 97). Regarding the educational level, 35% had a medium level and 35% had a high education level. With regard to lifestyle habits, almost half of all people enrolled had never smoked (45.2%, n = 112) and about one third followed a healthy diet, eating at least three portions of fruit and vegetables per day (30.7%, n = 76) (Table 1).

Most of the patients interviewed defined their state of health as “not good” (67.3%, n = 167) and just a third of the patients were hospitalised for surgical pathologies (34.3%, n = 85). Approximately half of the patients suffered from comorbidities (29.4% of patients were affected by two pathologies and 18.2% by three or more). Among the hospitalised patients, 63.3% (n = 157) had received advice about vaccination from their general practitioner and 47.0% (n = 117) from hospital healthcare workers. Almost half of the enrolled patients (46.8%, n = 116) reported that they had been vaccinated for influenza in the 2021–2022 influenza season. During the 2022–2023 influenza season, a total of 62.5% (n = 155) of patients agreed to be vaccinated in the University Hospital AUOP P. Giaccone at their discharge from hospitalisation (Table 2).

The comparison between patients who accepted vaccination and patients who remained unvaccinated for influenza during the 2022–2023 season showed that people who were vaccinated were more frequently those who had never smoked (52.9% vs. 32.3%; *p* = 0.019), had a low level of education (39.4% vs. 13.9%; *p* < 0.001) or were affected by clinical diseases (78.1% vs. 45.2%; *p* < 0.001) such as respiratory disease (29% vs. 15.1%, *p* < 0.001) and diabetes mellitus (36.2% vs. 21.3%; *p* = 0.019). On the other hand, the “unvaccinated” patients reported more frequently that they had not received advice about the vaccine, neither from their GP (47.3% vs. 30.3%, *p* = 0.026) or from hospital healthcare workers (71.0% vs. 32.5%; *p* < 0.001), and that they had not been vaccinated even during previous influenza seasons (73.1% vs. 41.3%; *p* < 0.001). Furthermore, considering patients suffering from oncological pathologies, a greater frequency of patients unvaccinated rather than vaccinated for influenza was observed (21.5% vs. 14.8%; *p* < 0.001) (Table 1 and Table 2).

Among the hospitalised patients, the factors significantly associated with the acceptance of influenza vaccination, from the multivariable analysis, were the following: to have received advice about vaccination from hospital healthcare workers (OR = 3.57; *p* = 0.001), to have been vaccinated against influenza during the previous season (OR = 3.16; *p* = 0.005) and to have a low educational level (OR = 3.56; *p* = 0.014) (Table 3).

## 4. Discussion

The suboptimal rate of influenza vaccine uptake among at-risk groups, despite the severity of influenza complications and the availability of vaccines, represents a serious public health challenge. The more traditional modalities for offering vaccination to elderly and frail people are not effective enough to reach the entire susceptible population. The need to evaluate alternative vaccine provision strategies led to the conduction of this study. 

The main finding was that actively offering influenza vaccination to hospitalised patients could be highly effective in increasing influenza vaccine uptake among the most vulnerable population. Our vaccination strategy allowed a 16% increase in influenza vaccination coverage in the same population compared to the previous season. Hospital-based opportunistic vaccination has also been shown to be very successful in other studies on influenza [19,20,21]. If the active offer of vaccination was carried out in every hospital or healthcare facility in Sicily, and was aimed at a higher number of inpatients, a significant increase in influenza vaccination coverage in the fragile population would be obtained. Moreover, the invitation to vaccinate patients for influenza during discharge should not be limited to a specific location but could be extended to all healthcare facilities in the Italian national territory. Currently, the Italian ministerial circular on the anti-influenza vaccination campaign does not promote the method of active offering to hospitalised patients [1]. Vaccination interventions that take place directly in a hospital setting offer several advantages: the possibility of intercepting the frail population who, due to chronic clinical conditions, make multiple visits to the hospital; detailed assessment of the health status of patients; removal of physical barriers related to access to immunisation services; and vaccine administration in a safe place [22]. Offering vaccination to a hospitalised patient is a strategy promoted by the World Health Organization to reduce “missed opportunities” for vaccination with the aim of improving the delivery of health services and promoting full synergy among healthcare professionals [23,24]. In order to enable vaccination during discharge, it is necessary to have a strategy for fully efficient collaboration between health professionals: public health physicians, hospital department physicians, nurses and health assistants [25]. The vaccination of frail patients should be part of the hospitalised patient management process: setting guidelines and protocols for the implementation of standing orders for offering vaccinations [20]. 

In accordance with the results of many studies that have reported improved uptake of the influenza vaccine following advice from healthcare workers, this study highlighted the crucial role of hospital healthcare workers in influenza vaccination acceptance [22,26,27]. For some individuals, especially patients with chronic or complex medical conditions, the opportunity to be informed about vaccines and to discuss vaccination safety and efficacy with a hospital specialist, who understands their drug regimen or their health status, is a critical factor for their vaccine uptake decision [22,26]. In detail, the analyses of this study identified the recommendation to vaccinate by a hospital healthcare worker, and not a primary care physician, as one of the strongest predictors of adherence to the influenza vaccine for the frail population. This important association is also supported by other authors who have explored the determinants for adherence to influenza vaccination in the hospital setting [28]. Future interventions should aim both at increasing awareness among healthcare professionals of their role to recommend influenza vaccination for patients with medical comorbidities and at promoting opportunistic hospital vaccination for this population. 

A higher uptake of vaccination during hospitalisation was observed for patients who reported also having received an influenza vaccination during the previous season. Several studies have showed that people who have already experienced vaccination have a greater propensity for and confidence in vaccines [27,29]. This finding could be the result of a general habit to get vaccinated after people have had a good experience with a previous vaccination, but it is more efficacious for immunisation against influenza because the annual vaccination of frail people is recommended. On the other hand, people who have never experienced the influenza vaccination have many perplexities about the safety and efficacy of the vaccine [30]. To combat “vaccine hesitancy” among those in frail categories, it could be very effective to improve clear and effective communication about influenza vaccinations and adopt alternative channels of information about immunisation practices that lead to the informed acceptance of vaccinations, such as vaccination counselling aimed at inpatients or outpatients and performed in the hospital. 

One of the main factors associated with influenza vaccine acceptance was a lower education level. This finding has already been explored in the literature, but the evidence is discordant [31,32,33]. Several studies correlated a higher level of education with greater knowledge and willingness to be immunised for influenza or, more generally, with a more marked propensity towards vaccination [34,35]. On the other hand, a Spanish study about the determinants of influenza vaccination in the over-65 population showed a high vaccine uptake in individuals without study qualifications or with a low education level [31]. In our setting, it is possible that people with a low level of education had not received information before the hospital counselling on influenza vaccination and therefore welcomed the doctors’ advice and adhered more to the vaccination recommendation. Conversely, people with a high education level, who had a low risk for being vaccinated in the hospital, were more sceptical about trusting or had lower confidence in influenza vaccination. Therefore, it is presumable that the most educated people received inaccurate information about health-related topics, such as immunisation, and they accessed misleading and unaccredited sources of information [32]. The crucial role of vaccination information sources has already been explored above [36]. Once again, the need for accurate information in the frail population about health and prevention practices, from reliable and accredited sources such as public health physician and hospital healthcare workers, is necessary. 

A lower adherence to influenza vaccination was observed among patients suffering from oncological pathologies compared to patients suffering from other pathologies, such as diabetes, cardiovascular diseases or respiratory diseases. Influenza virus infection in cancer patients during chemotherapy could result in suboptimal cancer treatment and cause delays in treatment, with possible consequences for malignant disease control [37]. Furthermore, influenza-related hospitalisation rates are four times higher and mortality is up to ten times higher among cancer patients compared with the general population [38]. Consequently, influenza vaccination is strongly recommended in cancer patients and must be repeated every year, with each new influenza season [39]. However, adherence to vaccination is low, probably due to a lack of data on the efficacy of influenza vaccination, on the optimal time interval for vaccine administration in relation to therapy or on the safety of vaccination [40]. Although there are not many studies evaluating the effectiveness of influenza vaccination among adult cancer-affected populations, there is evidence on the extent of the antibody response after influenza vaccination (seroconversion) which, although lower in adult cancer-affected populations than in the healthy population, determined there was a timely protective immunological response (seroprotection) for the majority of patients with solid tumours, regardless of the therapy in progress [37,40]. The ideal time to administer the vaccine to patients undergoing cancer treatment remains unclear. But, in accordance with the guidelines of the Italian Medical Oncology Association (AIOM), vaccination should be scheduled two weeks before the start of oncological therapies to try to avoid the phase of leukopenia induced by the therapies [39]. Current evidence supports the recommendation to offer influenza vaccination to cancer patients. It is essential to provide vaccination education. Healthcare workers should regularly recommend vaccination to patients in their clinical practice and, to this end, the strategy of actively offering influenza vaccination in hospital settings should help to dispel patients’ doubts and concerns. 

The main limitation of this study is the convenience sampling of hospitalised patients. The small sample size might not be representative of the frail population targeted for vaccination. Furthermore, influenza vaccination data for the previous 2021–2022 season were self-reported by the patients. Despite these limitations, this pilot study tested a vaccine-offer strategy aimed at the frail population that could be very effective in increasing influenza vaccination coverage.

## 5. Conclusions

Offering influenza vaccination to hospitalised patients during discharge could be a valid intervention and an effective strategy to increase immunisation coverage in the frail population. It might be very useful to define programs and protocols for the implementation of standing orders for the administration of vaccinations in the hospital setting. Furthermore, collaboration with hospital healthcare workers is essential to increase the adoption of influenza vaccination among frail people.

## Figures and Tables

**Table 1 vaccines-11-01829-t001:** Characteristics of enrolled patients and the differences between those vaccinated and unvaccinated for influenza.

General Information and Lifestyle						
		Total Respondents n (%)		Vaccinated n (%)	Unvaccinated n (%)	*p* Value
		248		155	93	
Age						
	<65 years old	119	48.0%	68 (43.9%)	51 (54.8%)	0.094
	≥65 years old	129	52.0%	87 (56.1%)	42 (45.2%)	
Sex						
	Male	109	43.9%	65 (41.9%)	44 (47.3%)	0.409
	Female	139	56.1%	90 (58.1%)	49 (52.7%)	
Body Mass Index						
	Underweight	4	1.6%	2 (1.3%)	2 (2.2%)	0.939
	Normal weight	100	40.3%	62 (40.0%)	38 (40.9%)	
	Overweight	90	36.3%	56 (36.1%)	34 (36.6%)	
	Obese	54	21.7%	35 (22.6%)	19 (20.4%)	
Educational level						
	Low	74	29.8%	61 (39.4%)	13 (13.9%)	<0.001
	Medium	87	35.1%	49 (31.6%)	38 (40.9%)	
	High	87	35.1%	45 (29.0%)	42 (45.2)	
Economic status						
	High	20	8.1%	7 (4.5%)	13 (13.9%)	0.100
	Medium–High	92	37.1%	58 (37.4%)	34 (36.6%)	
	Medium–Low	97	39.1%	66 (42.6%)	31 (33.3%)	
	Low	15	6.1%	9 (5.8%)	6 (6.5%)	
	Missing data	24	9.7%	15 (9.7%)	9 (9.7%)	
Smoker						
	Yes	61	24.6%	33 (21.3%)	28 (30.1%)	0.019
	No, never smoked	112	45.2%	82 (52.9%)	30 (32.3%)	
	No smoking, more than 1 year	48	19.4%	23 (14.8%)	25 (26.8%)	
	No smoking, less than 1 year	25	10.1%	16 (10.3%)	9 (9.7%)	
	Missing data	2	0.8%	1 (0.7%)	1 (1.1%)	
Daily fruit and vegetable intake						
	<3 portions a day	76	30.7%	54 (34.8%)	22 (23.7%)	0.081
	≥3 portions a day	166	66.9%	96 (61.9%)	70 (75.3%)	
	Missing data	6	2.4%	5 (3.2%)	1 (1.1%)	

**Table 2 vaccines-11-01829-t002:** Clinical conditions and vaccination attitude by influenza vaccination status.

		Total Respondents n (%)		Vaccinated n (%)	Unvaccinated n (%)	*p* Value
		248		155	93	
Health status						
	High	81	32.7%	48 (31.0%)	33 (35.5%)	0.463
	Low	167	67.3%	107 (69.0%)	60 (64.5%)	
Hospital ward						
	Clinical	163	65.7%	121 (78.1%)	42 (45.2%)	<0.001
	Surgical	85	34.3%	34 (21.9%)	51 (54.8%)	
Respiratory Diseases						
	Yes	59	23.8%	45 (29.0%)	14 (15.1%)	<0.001
	No	173	69.8%	107 (69.0%)	66 (70.9%)	
	m.d. *	16	6.4%	3 (1.9%)	13 (13.9%)	
Cardiovascular Diseases						
	Yes	70	28.2%	49 (31.6%)	21 (22.6%)	0.001
	No	162	65.3%	103 (66.5%)	59 (63.4%)	
	m.d. *	16	6.4%	3 (1.9%)	13 (13.9%)	
Diabetes Mellitus						
	Yes	72	29.0%	55 (35.5%)	17 (18.3%)	<0.001
	No	160	64.5%	97 (62.6%)	63 (67.7%)	
	m.d. *	16	6.4%	3 (1.9%)	13 (13.9%)	
Oncological Diseases						
	Yes	43	17.3%	23 (14.8%)	20 (21.5%)	
	No	189	76.2%	129 (83.2%)	60 (64.5%)	<0.001
	m.d. *	16	6.4%	3 (1.9%)	13 (13.9%)	
Comorbidity						
	1 disease	114	47.9%	66 (42.6%)	48 (51.6%)	<0.001
	2 diseases	73	29.4%	52 (33.5%)	21 (22.6%)	
	≥ 3 diseases	45	18.2%	34 (21.9%)	11 (11.8%)	
	m.d. *	16	6.5%	3 (1.9%)	13 (14.0%)	
Vaccinated for influenza during 2021–2022 season						
	Yes	116	46.8%	91 (58.7%)	25 (26.9%)	<0.001
	No	132	53.2%	64 (41.3%)	68 (73.1%)	
Vaccine advice from General Practitioner						
	Yes	157	63.3%	108 (69.7%)	49 (52.7%)	0.026
	No	91	36.7%	47 (30.3%)	44 (47.3%)	
Vaccine advice from Hospital Healthcare Workers						
	Yes	131	53.0%	104 (67.5%)	27 (29.0%)	<0.001
	No	117	47.0%	50 (32.5%)	66 (71.0%)	

m.d. * = missing data.

**Table 3 vaccines-11-01829-t003:** Univariable and multivariable analysis of factors associated with the acceptance of influenza vaccination.

		Crude OR [95% C.I.]	*p* Value	Adjusted OR [95% C.I.]	*p* Value
Sex					
	Female	Ref		Ref	
	Male	1.24 [0.74–2.08]	0.409	1.35 [0.68–2.85]	0.388
Age					
	<65	Ref		Ref	
	≥65	1.55 [0.93–2.61]	0.095	0.44 [0.19–1.01]	0.058
Economic Status					
	High	Ref		Ref	
	Medium–High	3.17 [1.15–8.71]	0.025	1.10 [0.31–3.85]	0.878
	Medium–Low	3.95 [1.44–10.8]	0.008	0.69 [0.18–2.68]	0.598
	Low	2.78 [0.70–11.1]	0.146	1.04 [0.21–5.27]	0.961
Educational Level					
	High	Ref		Ref	
	Medium	1.20 [0.66–2.18]	0.543	1.54 [0.65–3.36]	0.285
	Low	4.38 [2.11–9.10]	<0.001	3.56 [1.29–9.79]	0.014
Hospital Ward					
	Clinical	Ref		Ref	
	Surgical	0.23 [0.13–0.40]	<0.001	0.61 [0.28–1.34]	0.223
Comorbidity					
	1 disease	Ref		Ref	
	2 diseases	1.80 [0.96–3.37]	0.067	1.58 [0.73–3.46]	0.249
	≥3 diseases	2.25 [1.94–4.87]	0.040	1.89 [0.71–5.05]	0.201
Vaccine advice from General Practitioner					
	No	Ref		Ref	
	Yes	1.71 [1.04–2.79]	0.034	1.28 [0.62–2.65]	0.496
Vaccine advice from Hospital Healthcare Workers					
	No	Ref		Ref	
	Yes	4.02 [2.35–6.89]	<0.001	3.57 [1.65–7.75]	0.001
Vaccinated for Influenza during 2021–2022 season					
	No	Ref		Ref	
	Yes	5.01 [1.81–13.7]	0.002	3.16 [1.43–7.01]	0.005

## Data Availability

Data will be available upon request to the corresponding author.

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
