# Peer review of "Impact of Actively Offering Influenza Vaccination to Frail People during Hospitalisation: A Pilot Study in Italy"

_vaccines, 2023, doi:10.3390/vaccines11121829_

Round 1

Reviewer 1 Report

Comments and Suggestions for Authors

This study evaluates an intervention of offering influenza vaccination to frail patients at hospital discharge and to analyze the facilitators and barriers associated with influenza vaccination uptake.

 The study may be improved in some aspects related to the following suggestions.

 Line 22. Immunization has a typo.

 Line 30. The vaccination coverage in 2021-22 was 45.8% and in 2023-23 was 62.5%. The absolute increase in the coverage was 16 percent points, but the relative increase is (0.625-0.468)/0.468= 0.336 -> 33.6%.  Nevertheless, the expected coverage without intervention in the 2022-23 season would probably be higher than 45.8% due to the increasing coverage with the increase in age.

 Line 50-53. The authors say “According to WHO guidelines, a threshold of 75% influenza vaccination coverage is needed to contain the circulation of the virus among the general population, … (1) (8).” I am afraid that there are not conclusive evidence of that.  The COVID-19 pandemic have demonstrated that even with higher coverage, the control of the circulation of the virus is not possible. Among many explanations, as the influenza vaccine effectiveness usually reach coverages about 50% it will be difficult the control of transmission even with 100% vaccine coverage.

 Line 130. The term ‘majority’ may transmit the impression of a higher proportion than 56.1% or 52.0%, which are near half of the total.

 Table 2. Consider if percentages calculated by rows may be more informative given that vaccination in 2022-23 is the end-point (dependent variable).

The row named “Vaccinated for Influenza during 2021-2022 influenza season “ value “No” need to be revised.

 Line 175.  “The suboptimal rate of influenza vaccine uptake among risk groups…”

 Consider comment that patients with oncological diseases were vaccinated in lower percentage than other patients.

 The expected coverage without intervention in the 2022-23 season would probably be higher than 45.8% due to the increasing coverage with the increase in age. 

This intervention has a limited impact since only covers patients who were hospitalized during the vaccination campaign. They are a very small proportion of the target population for vaccination.

Author Response

This study evaluates an intervention of offering influenza vaccination to frail patients at hospital discharge and to analyze the facilitators and barriers associated with influenza vaccination uptake. The study may be improved in some aspects related to the following suggestions.

 Line 22. Immunization has a typo.

Thank you for your suggestion. “Despite the worldwide recommendations of influenza immunization, vaccination coverage for patients exposed to the highest risk of severe complications is still far from the optimal target “(Lines 23 – 24).

 Line 30. The vaccination coverage in 2021-22 was 45.8% and in 2023-23 was 62.5%. The absolute increase in the coverage was 16 percent points, but the relative increase is (0.625-0.468)/0.468= 0.336 -> 33.6%.  Nevertheless, the expected coverage without intervention in the 2022-23 season would probably be higher than 45.8% due to the increasing coverage with the increase in age.

As reported in the manuscript "Our vaccination strategy allowed a 16% increase in influenza vaccination coverage in the same population compared to the previous season" (Lines 194 - 195). The increase in age of just one year cannot justify a 16% increase in influenza vaccination uptake. Furthermore, Italian national data on influenza vaccination coverage show a slight decrease in the 2022-2023 season compared to the 2021-2022 season. Similarly, a decrease in vaccination coverage was also recorded for the Sicily region (https://www.salute.gov.it/portale/documentazione/p6_2_8_3_1.jsp?lingua=italiano&id=19).

 Line 50-53. The authors say “According to WHO guidelines, a threshold of 75% influenza vaccination coverage is needed to contain the circulation of the virus among the general population, … (1) (8).” I am afraid that there are not conclusive evidence of that.  The COVID-19 pandemic have demonstrated that even with higher coverage, the control of the circulation of the virus is not possible. Among many explanations, as the influenza vaccine effectiveness usually reach coverages about 50% it will be difficult the control of transmission even with 100% vaccine coverage.

Thank you for your suggestion. The revised manuscript faithfully reports what is recommended by the guidelines and the Italian Ministry of Health (Lines 53 - 57).

 Line 130. The term ‘majority’ may transmit the impression of a higher proportion than 56.1% or 52.0%, which are near half of the total.

Thank you for the revision. “Almost half of the people interviewed were female (56.1%, n=139) and over 65 years old (52.0%, n=129)” (Lines 139-140).

 Table 2. Consider if percentages calculated by rows may be more informative given that vaccination in 2022-23 is the end-point (dependent variable).

We added the confidence interval for each OR of the univariate and multivariate analysis.

The row named “Vaccinated for Influenza during 2021-2022 influenza season “ value “No” need to be revised.

Thanks for your suggestion, it was a formatting error (Table 2).

 Line 175.  “The suboptimal rate of influenza vaccine uptake among risk groups…”

Thank you for your suggestion. It was just a typo.

 Consider comment that patients with oncological diseases were vaccinated in lower percentage than other patients.

Thanks for the valuable suggestion. We have expanded the discussion section with arguments and proposals regarding the vaccination of cancer patients (Lines 262-286).

 The expected coverage without intervention in the 2022-23 season would probably be higher than 45.8% due to the increasing coverage with the increase in age. 

The increase in age of just one year cannot justify a 16% increase in influenza vaccination uptake. Furthermore, Italian national data on influenza vaccination coverage show a slight decrease in the 2022-2023 season compared to the 2021-2022 season. Similarly, a decrease in vaccination coverage was also recorded for the Sicily region. (https://www.salute.gov.it/portale/documentazione/p6_2_8_3_1.jsp?lingua=italiano&id=19).

This intervention has a limited impact since only covers patients who were hospitalized during the vaccination campaign. They are a very small proportion of the target population for vaccination.

The influenza vaccine is administered during the vaccination campaign, which generally begins at the end of October and ends in February. Our intervention offering influenza vaccination was aimed at the hospitalized population during the months of the vaccination campaign. It is not to be considered a method of vaccination offering that must replace the more traditional offering by general practitioners, but a vaccination strategy useful for increasing vaccination coverage among the frail population. Furthermore, despite the small sample size reported within the limits of the manuscript, the results obtained show statistical significance.

Reviewer 2 Report

Comments and Suggestions for Authors

This study investigated factors associated with influenza vaccine acceptance and uptake among frail people at discharge, which may provide some evidence for improving influenza vaccine uptake in this high-risk population. However, there are a few issues that need to be addressed.

1.       The use of “effectiveness” in the title and the whole manuscript is confusing since this study did not assess vaccine effectiveness. Suggest replacing the term “effectiveness” with “impact” or “effect”, etc.

2.        My understanding is that study participants were recommended and given influenza vaccination at discharge from hospitals, not during the whole period of hospitalization. Suggest describing this more clearly through the manuscript.

3.       Line 50 - Please write the term “WHO” in full when it occurred first in the manuscript.

4.       The definition of frail people can vary from study to study, therefore it’s important to clearly define it in the study.

5.       After data were collected via questionnaires, what was the data entry process? Please supplement this in the methods section.

6.       It’s essential to report the response rate of a survey. Please add this information in the results section.

7.       What variables were included in the multivariate models, please specify.

8.       Please provide 95% confidence intervals of adjusted ORs.

9.       Line 205 “This study highlighted in according to literature” what does this mean? Please rewrite the whole sentence.

Comments on the Quality of English Language

10.   The manuscript needs language corrections before being published.

11.   There are some typos and grammatical errors, including

·         Line 22 replace “immunizzation” with “immunization”

·         Line 98 replace “all eligible patient” with “all eligible patients”

·         Line 99 replace “the aim of identify” with "the aim of identifying”

Author Response

Response to Reviewer 2 Comments

  1.  Thank you for your suggestion, we preferred to replace the term "effectiveness" with "impact".
  2. “The vaccine was offered directly during discharge from the hospital, after informing patients about the risks and benefits of influenza vaccine and after assessing the patient's clinical condition suitable to receive the vaccination (absence of inflammation and fever)” (Lines 101 – 104). We tried to explain the way vaccination was offered at discharge. Thank you for the suggestion.
  3. Thanks for the review. “According to World Health Organization guidelines, a threshold of 75% vaccination coverage against influenza is necessary among the general population, as a minimum achievable objective, and a threshold of 95% among the frail population to prevent possible negative outcomes and reduce the morbidity related to seasonal influenza” (Lines 53-57).
  4. In order to describe the study population more in detail, we added the description of the frail population, that is the primary target of influenza vaccination. “Influenza vaccination was offered to frail patients, i.e., over 65 years old or affected by chronic clinical conditions or comorbidities, the main target of vaccination stipulated by ministerial recommendations for the influenza campaign” (Lines 98 – 101)..
  5. Thanks for your suggestion. “Patient information was transferred from paper formats to computer files; a database was constructed to record data using Excel-office 2021” (Lines 121 – 122).
  6. “… 253 hospitalized patients during the 2022-2023 influenza season were enrolled, but 5 people denied consent to be interviewed (response rate 98%)” (Lines 138 – 139).
  7. “A multivariable logistic regression model was built to analyze factors associated at univariable analysis with a P-value lower or equal than 0.05. Moreover a priori confounding variables, such as age and sex, were included in the multivariable model. For all analyses, a P-value of 0.05 was assumed to be statistically significant” (Material and methods). The construction of the multivariate model was reported in the "Materials and methods" section of the manuscript and the variables included are reported in Tab 3 "Univariable and multivariable analysis of factors associated with acceptance of influenza vaccination".
  8. Thanks to your advice, we have provided confidence intervals for univariate and multivariate analyses of factors associated with acceptance of influenza vaccination (Table 3).
  9. We rewrote the sentence to make it easier to understand: “In accordance with the results of many studies that have reported improved uptake of the Influenza vaccine following advice from healthcare workers, this study highlighted the crucial role of hospital health workers in the influenza vaccination acceptance [22,26,28]” (Lines 217 - 219).

Comments on the Quality of English Language

As you requested, the manuscript underwent extensive English revisions. Thank you for your suggestions.
